# Does public transport use prevent declines in walking speed among older adults living in England? A prospective cohort study

Patrick Rouxel,[1] Elizabeth Webb,[2] Tarani Chandola[3]

[1]CLOSER, Department of Social Science, University College London Institute of Education, London, UK
[2]Department of Epidemiology and Public Health, International Centre for Lifecourse Studies, University College London, London, UK
[3]CMIST and Department of Social Statistics, University of Manchester, London, UK

**Correspondence to**
Dr Patrick Rouxel;
patrick.rouxel@ucl.ac.uk

## ABSTRACT

**Objectives** Although there is some evidence that public transport use confers public health benefits, the evidence is limited by cross-sectional study designs and health-related confounding factors. This study examines the effect of public transport use on changes in walking speed among older adults living in England, comparing frequent users of public transport to their peers who did not use public transport because of structural barriers (poor public transport infrastructure) or through choice.

**Design** Prospective cohort study.

**Setting** England, UK.

**Participants** Older adults aged ≥60 years eligible for the walking speed test. 6246 individuals at wave 2 (2004–2005); 5909 individuals at wave 3 (2006–2007); 7321 individuals at wave 4 (2008–2009); 7535 individuals at wave 5 (2010–2011) and 7664 individuals at wave 6 (2012–2013) of the English Longitudinal Study of Ageing.

**Main outcome measure** The walking speed was estimated from the time taken to walk 2.4 m. Fixed effects models and growth curve models were used to examine the associations between public transport use and walking speed.

**Results** Older adults who did not use public transport through choice or because of structural reasons had slower walking speeds (−0.02 m/s (95% CI −0.03 to −0.003) and −0.02 m/s (95% CI −0.03 to −0.01), respectively) and took an extra 0.07 s to walk 2.4 m compared with their peers who used public transport frequently. The age-related trajectories of decline in walking speed were slower for frequent users of public transport compared with non-users.

**Conclusions** Frequent use of public transport may prevent age-related decline in physical capability by promoting physical activity and lower limb muscle strength among older adults. The association between public transport use and slower decline in walking speed among older adults is unlikely to be confounded by health-related selection factors. Improving access to good quality public transport could improve the health of older adults.

## INTRODUCTION

Declines in walking speed and grip strength are markers of ageing and are associated with all-cause mortality[1] and poorer health in older populations.[2] Maintaining physical

### Strengths and limitations of the study

► Previous cross-sectional research on the protective role of public transport use in relation to age-related functional declines may have been biased by health selection processes as people age.

► Older adults with deteriorating health are less likely to use public transport and their poor health could determine their functional decline, rather than their lack of public transport use.

► This longitudinal study of over 7000 adults living in England suggests that the inference that frequent use of public transport may prevent age-related decline in physical capability is robust to such potential biases.

► While this is not a causal analysis, we have controlled for within-person and between-person factors that could bias the association between public transport use and walking speed decline.

► Given the current context of cuts in public transport availability in England, this research suggests that such cuts may result in faster declines in physical functioning as people age.

capability is a prerequisite for older people to engage independently in many social activities[3] and for reducing social exclusion.[4 5]

Functional capacity and muscle strength are key dimensions of sarcopaenia.[6] Compelling evidence supports the efficacy of physical activity in maintaining muscle mass, strength and function in older adults.[7 8] Older adults may add regular physical activity into daily life by walking and maintaining balance on moving vehicles such as buses or trains.[7] Public transport-related physical activity was associated with a larger reduction in mortality for older adults (>70 years) compared with younger adults.[9] Public transport users are more physically active than non-users of public transport.[10–12]

Disability is not the only barrier to the use of public transport for many older people. Other barriers include the costs and poor

quality of public transport.[5 13 14] These barriers, in turn, suppress leisure, social interactions and shopping activities.[13 15] Accessible, affordable and convenient transport is important to enable older people to access services and amenities. It is particularly important to consider such barriers to public transport use given the current context of cuts to local bus services in England.[16]

Despite concessionary bus passes in the UK offering free bus travel to those over the State Pension Age, in England a third of older adults aged >65 never use public transport, whereas another third use it very infrequently.[17] Evidence from the English Longitudinal Study of Ageing (ELSA)[18 19] and other studies[5 20] showed that free bus travel for older people was associated with increased active travel and raised physical activity. A review has reported between 8 and 33 min of additional physical activity by walking through use of public transport, although most of the studies included were not focused on older adults.[11] Although walking speed declines with age,[21 22] there is some cross-sectional evidence that older women with a free bus pass use public transport more often and have faster walking speeds than those who do not hold a bus pass.[18] Moreover, public transport use is associated with lower levels of obesity and may have a protective effect against becoming obese.[19] A recent longitudinal study on the use of active or public transport versus cars to commute to work showed that people who changed from active or public commuting to car commuting had an increase in body mass index (BMI) of 0.3 kg/m$^2$, whereas those who changed from car commuting to active or public commuting had a decrease in BMI of 0.3 kg/m$^2$.[23]

However, the key limitation in the existing evidence is that negative health selection is not taken into account. Older adults with deteriorating health may be less likely to use public transport and their poor health could determine the decline in walking speed, rather than their lack of public transport use. There is also a lack of analyses on different reasons for not using public transport. Older adults may not use public transport because of health problems and disabilities[3 24] or because reliable public transport is not available[13] or because they prefer using their own vehicles.[25] Separating out these reasons for not using public transport is important: if a negative effect of not using public transport on walking speed is observed among older adults who do not use public transport due to structural reasons (poor public transport infrastructure) or through choice, this suggests that the observed association between public transport use and walking speed is not confounded by health.

An additional test of the specificity of the public transport–walking speed association is whether similar associations are observed between public transport use and upper body strength as measured by grip strength. As moderate to vigorous physical activity, including public transport use, does not increase grip strength,[26] any association between public transport use and grip strength could be caused by confounding factors such as stronger people being selected into using public transport. Moreover, if public transport use affects walking speed, the mechanisms are likely to be through walking-related physical activity and slower declines in lower limb strength.

Our study will address the following research questions:

RQ1: Do older adults who frequently use public transport have faster walking speeds than those who do not use public transport for structural reasons or through choice? Is some of the association between public transport use and walking speed going through the mechanisms of physical activity and lower limb strength?

RQ2: Is the association of public transport use with muscle function deficit specific to lower limb muscle strength? Does public transport use also predict stronger grip strength?

RQ3: Are declines of walking speed with age slower for older adults who use public transport often, compared with their peers who do not use public transport for structural reasons or through choice?

## METHODS
### Data
The data come from waves 2 to 6 of the ELSA, where individuals aged ≥50 years living in private households in England were followed and reinterviewed every 2 years.[27] The ELSA sample was refreshed at waves 3, 4 and 6 to ensure the sample remained representative of the population aged ≥50 years. The National Research Ethics Service approved the study, and all participants gave their informed consent. We used data from older adults aged ≥60 years—those who were eligible for the walking speed test—consisting of 6246 individuals at wave 2 (2004–2005); 5909 individuals at wave 3 (2006–2007); 7321 individuals at wave 4 (2008–2009); 7535 individuals at wave 5 (2010–2011) and 7664 individuals at wave 6 (2012–2013). Data from wave 1 were omitted, since the reasons why people did not use public transport were not asked. Data were collected though face-to-face interview and a self-completion questionnaire. In addition, there was a nurse visit at waves 2, 4 and 6. The ELSA data and documentation are publicly available from the UK Data Service (http://www.esds.ac.uk/findingData/snDescription.asp?sn=5050).

### Variables
Walking speed (m/s) was measured among participants aged ≥60 years at every ELSA wave. They were asked to twice walk a distance of 8 feet (2.4 m) at their usual pace. Walking aids were permitted. We used the mean walking speed based on the two timings.

Grip strength (kg) was used as an indicator of upper body strength. Participants were asked to squeeze a hand-held dynamometer up to three times with each hand in waves 2, 4 and 6. We used the mean of the three measurements for the dominant hand.

Chair stands were used as an indicator of lower limb muscle strength. Respondents at waves 2, 4 and 6 aged

<70 years were asked to do 10 chair stands and those aged 70+ were asked to do 5 chair stands. This was grouped into people who could not complete either 5 or 10 chair stands (including those who could not complete a single chair stand), those who took longer than the median time to complete 5/10 chair stands, and those who completed the task in less than the median time.

Participants were asked how often they used public transport. In addition, those who rarely or never used public transport were asked to provide the reasons of not using public transport more often. We then derived the frequency variable as follows: every day/nearly every day; two to three times a week; once a week; no use because did not need to (or in other words through 'choice'); no use because of health problems and no use because of structural reasons. Structural reasons, for these purposes, include: not convenient, does not go where they want, infrequent, unreliable, too expensive, too dirty and fear of crime. At wave 2, the frequency of use variable had different responses which we mapped onto the later wave responses as follows: a lot=nearly every day; quite often=2–3 times a week; sometimes=once a week and rarely/never=no use.[19]

### Covariates

A quadratic term for continuous age was specified to characterise non-linear age effects.

Other covariates included gender; marital status (married, divorced/separated, widowed, never married); cohabiting status (currently living with a partner or not) and urban/rural areas (urban, town, village, hamlet)—the rural–urban definition is applied to the Census Output Area that each individual lives in; quintiles of non-pension wealth; access to car (whether driver or passenger); employment status (employed, retired, other); National Statistics Socio-economic Classification (NS-SEC) social class; and smoking (never, former, current). Mobility difficulties were assessed by asking participants whether they had difficulties with 10 common functions (eg, walking 100 yards, climbing several flights of stairs without resting, climbing one flight of stairs without resting, lifting or carrying weights). We derived a variable with three categories: no difficulties; one to three difficulties, four or more difficulties. The number of functional limitations in activities of daily living (ADLs) provides an indication of disability. The ADLs scale[28] comprises problems with dressing, walking, bathing, eating (such as cutting up food), getting out of bed and using the toilet. We derived a binary variable no limitations in ADLs; at least one limitation in ADLs. Depressive symptoms were measured using the eight-item version of the Center for Epidemiologic Study Depression (CES-D) scale.[29] Participants were asked how often they participated in mild, moderate or vigorous physical activity including participation in occupations that involve physical work. Based on their responses, they were classified into the following categories: sedentary; low; moderate or high level of physical activity.[30] Cognitive function was assessed through two memory tests.

Study participants were asked to recall a list of 10 words immediately after reading them and then again after a 5 min delay. We computed an overall memory score (range, 0–20) using both the immediate and delayed recall results (between-test correlation coefficient=0.70). Orientation in time (day, month, year and day of the week) is another test of memory.

### Analysis

For RQs 1 and 2, fixed effects linear and multinomial regression models were used to regress walking speed, physical activity, chair stands and grip strength on the frequency of using public transport. These models investigate the effects of within individual (time varying) changes in public transport use on changes in walking speed, physical activity, lower limb and upper body muscle strength, taking into account other time varying covariates. We also used fixed effects multinomial logit models to examine whether the direction of association could be in the opposite direction, that is, whether changes in the use/non-use of public transport (for different reasons) were predicted by changes in walking speed and other covariates. We estimated these models in Stata V.14.0.

For RQ3, we used multilevel (random effects) growth curve models to estimate age-related trajectories of walking speed for different categories of public transport use. These models can be used to describe how different trajectories of walking speed change with age, by interacting age with the frequency of public transport use. Individual trajectories (at level 2) of walking speed (at level 1) are estimated, with a random slope of age (at level 2) characterising differences in individual trajectories of walking speed. As participants with a single walking speed measurement can contribute to the overall growth curve model, we additionally estimated age-related trajectories of walking speed for those participants with at least three waves of walking speed measurements. We estimated these models in MLwiN V.2.1.

### Missing data

For the walking speed analyses, there are 29 894 observations of walking speed between waves 2 and 6, which reduced to 27 525 observations in the statistical models due to missing data in the covariates. Attrition between waves was not strictly monotonic—some ELSA participants returned to the study after missing one or two waves of data collection. Hence, rather than analysing factors only associated with attrition, we analysed factors associated with missing data in any of the independent and dependent variables and covariates, conditional on observation of a participant's walking speed at baseline.

Analyses of the pattern of missingness in the cohort with a baseline walking speed measurement revealed that 33% of that cohort had subsequently dropped out by wave 6, 16% were missing a walking speed measure at wave 6, 5% were missing both a walking speed and wealth measurement and 3% were missing a wealth measurement. Other covariates accounted for <2% of the missing data.

We modelled the odds of having any missing data (conditional on having a baseline walking speed measurement) as shown in the online supplementary table S1. Women and older participants were less likely to have any missing data, especially older women participants. Participants who did not use public transport because of health problems were more likely to be missing compared with those who used public transport frequently, but those who did not use public transport for other reasons were not more likely to be missing. Socioeconomic disadvantage (not having access to a car/van and being in the semi-routine and routine occupational class) was associated with higher odds of being missing, as was having a disability, low physical activity, low memory scores and higher levels of depressive symptoms.

The ELSA study team provides longitudinal weights for the core ELSA members present at each wave from the first wave. We did not use these for our analyses as, since they are not available for the ELSA refreshment sample members, using the longitudinal weights would have reduced our sample size by more than half. Instead, we used the wave specific cross-sectional weights in both the fixed effects and multilevel growth curve models, in order to make our analyses representative of non-institutionalised older adults living in England. These cross-sectional weights take into account the greater likelihood of non-response by participants who have poorer health and who are more socioeconomically disadvantaged.[31]

## RESULTS

Table 1 shows the distribution of all the variables used in the analysis. The mean walking speed was 0.85 m/s and the mean age was 70.5 years. Thirty-four per cent of ELSA respondents across waves 2–6 reported taking public transport at least once a week, whereas 33% reported not using public transport because of structural reasons. Most of the sample (73%) lived in urban areas, and almost 60% reported at least one difficulty with mobility.

Table 2 reports the mean walking speed by frequency of public transport use at waves 2–6. Those who did not use public transport due to health problems had the slowest walking speed, whereas those who did not use public transport because of structural reasons had the fastest walking speed. There was no pattern of slower walking speed in later waves due to refreshment samples which resulted in the mean age at wave 6 being younger than at wave 2.

Table 3 reports the results of the fixed effects and multilevel growth curve (random effects) models of walking speed—the full models are shown in online supplementary table S2 and S3. In the fixed effects model, the linear and quadratic terms of age were negative, suggesting that as age increased, walking speed declined faster. Compared with the reference group who used public transport nearly every day, the coefficients for all the other frequency groups were negative, suggesting that using public transport nearly every day had a protective effect on walking speed. Those who did not use public transport because of health problems had the biggest decline in walking speed (−0.06 m/s), but those who did not use public transport because they did not need to, or because of structural reasons were also more likely to have a decline in walking speed (−0.02 m/s). A difference of 0.02 m/s is an extra 0.07 s taken to do the walking speed test. The interaction between age and frequency of use of public transport was not significant in the fixed effects model.

We additionally examined whether the association between public transport use and walking speed decreased when taking into account lower limb muscle strength (chair stands) and the interaction between physical activity and age (online supplementary table S4). The coefficients in the model without controlling for these potential mechanisms for those who did not use public transport because they did not need to or because of structural reasons was −0.02 m/s; although due to the smaller sample size (chair stands were only collected at waves 2, 4 and 6), the 95% CIs overlapped 0. Once chair stands and the interaction between physical activity and age were controlled for, these coefficients reduced by about half to −0.01, suggesting that some of the association between public transport use and walking speed among older adults is statistically explained by lower limb muscle strength and physical activity.

We also used fixed effects models to examine whether changes in the use/non-use of public transport (for different reasons) were predicted by walking speed and the other covariates (online supplementary table S5). For this analysis, we grouped the frequency variable (the dependent variable) into fewer groups: (1) used public transport, the reference category; (2) did not use because of no need; (3) did not use because of health problems and (4) did not use because of structural reasons. Slower walking speed and poorer health did not predict changes in the use/non-use of public transport because of the lack of need or structural reasons, but slower respondents were more likely not to use public transport for health reasons. Respondents with access to a car/van were less likely to use public transport. Respondents with mobility difficulties, low physical activity levels and higher levels of depressive symptoms were more likely to not use public transport for health reasons. Living in an isolated area increased the likelihood of a person reporting not using public transport because of structural reasons.

The results of the fixed effects model predicting grip strength are shown in the online supplementary table S6. Unsurprisingly, older adults who did not use public transport due to health problems had weaker grip strength than those who used public transport nearly/every day. However, there were no differences in grip strength between the latter group and those who did not use public transport due to structural reasons or because they did not need to. In contrast, not using public transport because of structural barriers was associated with decrements in lower limb muscle strength. Such ELSA participants were more likely to become unable to complete the

**Table 1** Distribution (percentage/mean) of all the variables in the analysis, observations (n) across ELSA waves 2–6 among respondents with walking speed data

| Variables | %/Mean (SD) | Observations (n) across six waves | Variables | %/Mean (SD) | Observations (n) across six waves |
|---|---|---|---|---|---|
| Walking speed (m/s) | 0.9 (0.3) | 29894 | Urban/rural | | |
| Chair stands | | | Urban | 72.6% | 21679 |
| Could not complete test | 17.5% | 2780 | Town | 12.6% | 3771 |
| Completed test slower | 45.8% | 7288 | Village | 10.8% | 3213 |
| Completed test faster | 36.8% | 5857 | Hamlet | 4.0% | 1198 |
| Frequency of public transport use | | | Marital status | | |
| Every day or nearly every day | 8.5% | 2540 | Married | 65.0% | 19429 |
| Two or three times a week | 14.4% | 4300 | Separated/divorced | 9.8% | 2923 |
| Once a week | 11.2% | 3342 | Widowed | 20.5% | 6117 |
| No use: no need | 27.9% | 8337 | Never married | 4.8% | 1421 |
| No use: health problems | 5.5% | 1631 | Mobility difficulties | | |
| No use: structural reasons | 32.6% | 9741 | None | 40.7% | 12155 |
| **Age** | 70.5 (7.7) | 29894 | 1 to 3 | 37.6% | 11233 |
| Gender | | | ≥4 | 21.8% | 6502 |
| Man | 45.9% | 13728 | Functional status (ADLs) | | |
| Woman | 54.1% | 16166 | No limitation | 82.1% | 24538 |
| Wealth quintiles | | | At least 1 limitation | 17.9% | 5355 |
| Poorest | 16.4% | 4660 | **CES-D depression score** | 1.3 (1.8) | 29579 |
| 2nd | 18.8% | 5327 | **Physical activity** | | |
| 3rd | 21.2% | 5999 | Sedentary | 5.0% | 1503 |
| fourth | 21.6% | 6135 | Low | 26.1% | 7770 |
| Richest | 22.0% | 6245 | Moderate | 51.1% | 15236 |
| Employment status | | | High | 17.8% | 5321 |
| Employed | 16.8% | 5010 | Smoking status | | |
| Retired | 74.0% | 22081 | Never smoked | 36.6% | 10931 |
| Other | 9.2% | 2748 | Ex-smoker | 52.1% | 15551 |
| Social class | | | Current smoker | 11.2% | 3351 |
| Managerial | 32.2% | 9458 | Date/Day orientation | | |
| Intermediate | 14.0% | 4093 | All dates/day incorrect | 0.7% | 198 |
| Self-employed | 11.7% | 3434 | 3 incorrect | 0.6% | 192 |
| Lower supervisory | 10.5% | 3083 | 2 incorrect | 1.9% | 579 |
| Semi-routine | 31.6% | 9270 | 1 incorrect | 17.9% | 5357 |
| Cohabiting status | | | All dates/day correct | 78.8% | 23548 |
| Living alone | 33.2% | 9909 | Access to car | | |
| Living with partner | 66.9% | 19985 | Yes access to car | 83.4% | 24944 |
| Memory test (n of words) | 9.9 (3.6) | 29833 | No access to car | 16.6% | 4950 |

ADLs, activities of daily living; CES-D, Center for Epidemiologic Study; ELSA, English Longitudinal Study of Ageing.

chair stand test, relative to those who completed the test quicker (online supplementary table S7). Furthermore, older ELSA respondents who did not use public transport because they did not need to or because of structural barriers were also more likely to become sedentary relative to those engaging in high physical activity (online supplementary table S8).

Turning to RQ3, in the first growth curve model (table 3, column 3), we see similar estimates for the intercept and age coefficients compared with the fixed effects model.

**Table 2** Weighted mean (95% CI) of walking speed (m/s) by frequency of public transport use at waves 2–6 of ELSA

|  | Every day or nearly every day | Two or three times a week | Once a week | No use: no need | No use: health problems | No use: structural reasons |
|---|---|---|---|---|---|---|
| Wave 2 | 0.79 (0.77 to 0.81) | 0.79 (0.77 to 0.81) | 0.85 (0.84 to 0.87) | 0.84 (0.83 to 0.86) | 0.45 (0.42 to 0.47) | 0.90 (0.89 to 0.92) |
| n | 652 | 695 | 1071 | 1475 | 333 | 1252 |
| Wave 3 | 0.77 (0.75 to 0.80) | 0.79 (0.77 to 0.80) | 0.79 (0.76 to 0.81) | 0.84 (0.82 to 0.85) | 0.44 (0.42 to 0.47) | 0.90 (0.88 to 0.91) |
| n | 389 | 694 | 462 | 1431 | 289 | 1863 |
| Wave 4 | 0.78 (0.76 to 0.81) | 0.80 (0.78 to 0.82) | 0.85 (0.83 to 0.88) | 0.84 (0.83 to 0.85) | 0.47 (0.44 to 0.5) | 0.89 (0.88 to 0.91) |
| n | 499 | 956 | 556 | 1584 | 295 | 2157 |
| Wave 5 | 0.81 (0.78 to 0.83) | 0.84 (0.83 to 0.86) | 0.86 (0.84 to 0.88) | 0.88 (0.87 to 0.89) | 0.47 (0.44 to 0.49) | 0.91 (0.90 to 0.93) |
| n | 472 | 935 | 624 | 1610 | 315 | 2148 |
| Wave 6 | 0.81 (0.79 to 0.84) | 0.84 (0.83 to 0.86) | 0.84 (0.82 to 0.87) | 0.90 (0.88 to 0.91) | 0.49 (0.46 to 0.51) | 0.93 (0.91 to 0.94) |
| n | 450 | 906 | 547 | 1884 | 346 | 1881 |

ELSA, English Longitudinal Study of Ageing.

**Table 3** Selected coefficients (95% CI) from the fixed effects and growth curve models of walking speed (m/s), ELSA waves 2–6

|  | Fixed effects model | Growth curve model 1 | Growth curve model 2 |
|---|---|---|---|
| **Fixed part** | | | |
| *Intercept* | **0.820 (0.756 to 0.884)** | **0.823 (0.779 to 0.866)** | **0.830 (0.791 to 0.87)** |
| *Age-centred (linear term)* | **−0.008 (−0.01 to −0.007)** | **−0.007 (−0.008 to −0.006)** | **−0.006 (−0.007 to −0.004)** |
| *Age (quadratic term)* | **−0.0003 (−0.0004 to −0.0003)** | **−0.0002 (−0.0002 to −0.0001)** | **−0.0002 (−0.0002 to −0.0002)** |
| *p, 2df* | *<0.001* | *<0.001* | *<0.001* |
| Frequency of public transport use(ref: every day or nearly every day) | | | |
| Two or three times a week | −0.012 (−0.024 to 0.0001) | −0.006 (−0.016 to 0.003) | −0.005 (−0.018 to 0.008) |
| Once a week | **−0.020 (−0.034 to −0.005)** | **−0.011 (−0.022 to −0.0002)** | −0.006 (−0.019 to 0.008) |
| No use: no need | **−0.018 (−0.032 to −0.003)** | **−0.011 (−0.022 to −0.001)** | −0.005 (−0.018 to 0.008) |
| No use: health problems | **−0.058 (−0.075 to −0.040)** | **−0.090 (−0.104 to −0.077)** | **−0.115 (−0.135 to −0.095)** |
| No use: structural reasons | **−0.020 (−0.035 to −0.006)** | **−0.009 (−0.019 to 0.002)** | −0.002 (−0.014 to 0.011) |
| *p, 5df* | | *<0.001* | *<0.001* |
| Interaction between age (linear term) and frequency of public transport use | | | |
| Age* two or three times a week | | | −0.0003 (−0.002 to 0.001) |
| Age* once a week | | | −0.001 (−0.002 to 0.0004) |
| Age* no use: no need | | | **−0.001 (−0.002 to −0.00002)** |
| Age* no use: health problems | | | **0.002 (0.0001 to 0.003)** |
| Age* no use: structural reasons | | | **−0.001 (−0.003 to −0.0002)** |
| *p, 5df* | | | *<0.001* |
| **Random part** | | | |
| *Level 2 (individual)* | | | |
| Intercept variance | | 0.0238 | 0.0238 |
| Age-centred (linear term) variance | | 0.00002 | 0.00002 |
| Covariance of intercept and age centred | | −0.0004 | −0.0004 |
| *Level 1 (wave)* | | | |
| Intercept | | 0.020 | 0.020 |
| N observations (level 1) | 27 509 | 27 509 | 27 509 |
| N clusters (level 2) | 9656 | 9656 | 9656 |
| Goodness of fit | Adj R-sq: 0.7273 | Deviance: −13 719.59 | Deviance: −13 746.54 |

Bold values indicate statistical significance (p<0.05).
ELSA, English Longitudinal Study of Ageing.

All the coefficients for the frequency variable were also negative, although the negative effect of not using public transport due to structural reasons on walking speed was small and not different from those who used public transport nearly every day (the reference group). The second growth curve model added in the interaction between the frequency and age. With increasing age, the effect on walking speed of not using public transport for structural reasons or because the respondent did not feel the need to became increasingly more negative. The trajectories of these three groups (those who used public transport nearly every day, those who did not use because of structural reasons, and those who did not use because they did not need to) are shown in figures 1 and 2. In both figures, the decline in walking speed with age started to diverge around age 75, when there was a slower decline in walking speed for those who used public transport nearly every day, and a much steeper decline for those who did not use public transport because of structural reasons or a lack of need. The upper 95% CIs of the latter two groups clearly did not overlap with the estimated trajectories of the frequent public transport users after about the age of 75. We also estimated the growth curve model for those participants with at least three waves of walking speed measurements (supplementary table S9) and found very similar estimates to the sample including all ELSA participants with at least one walking speed measurement (online supplementary table S3).

## DISCUSSION

This study found evidence that older adults living in England who frequently used public transport had faster walking speeds than their peers who did not use public transport. Results from fixed effects and multilevel growth curve models showed similar patterns. In fixed effects models, frequent public transport use among older adults had a protective effect on walking speed. Unsurprisingly, not using public transport due to health reasons had the largest negative effect on walking speed. However, not using public transport due to other reasons also had a negative effect on walking speed. While the effect size of 0.02 m/s associated with not using public transport due to structural reasons may appear small, the predicted levels of walking speed in this cohort of older adults were well below the recommended 1.2 m/s walking speed needed for standard pedestrian crossings.[32] Any increase in the walking speed of older adults through factors such as physical activity and increased public transport use may help them cross the road safely.

The results of the fixed effects models were corroborated by the trajectories of walking speed decline shown in the growth curve models. In the growth curve models, older adults who did not use public transport due to structural reasons or because of a lack of need ('through choice') had a faster decline in walking speed after the age of 75 than those who used public transport nearly every day.

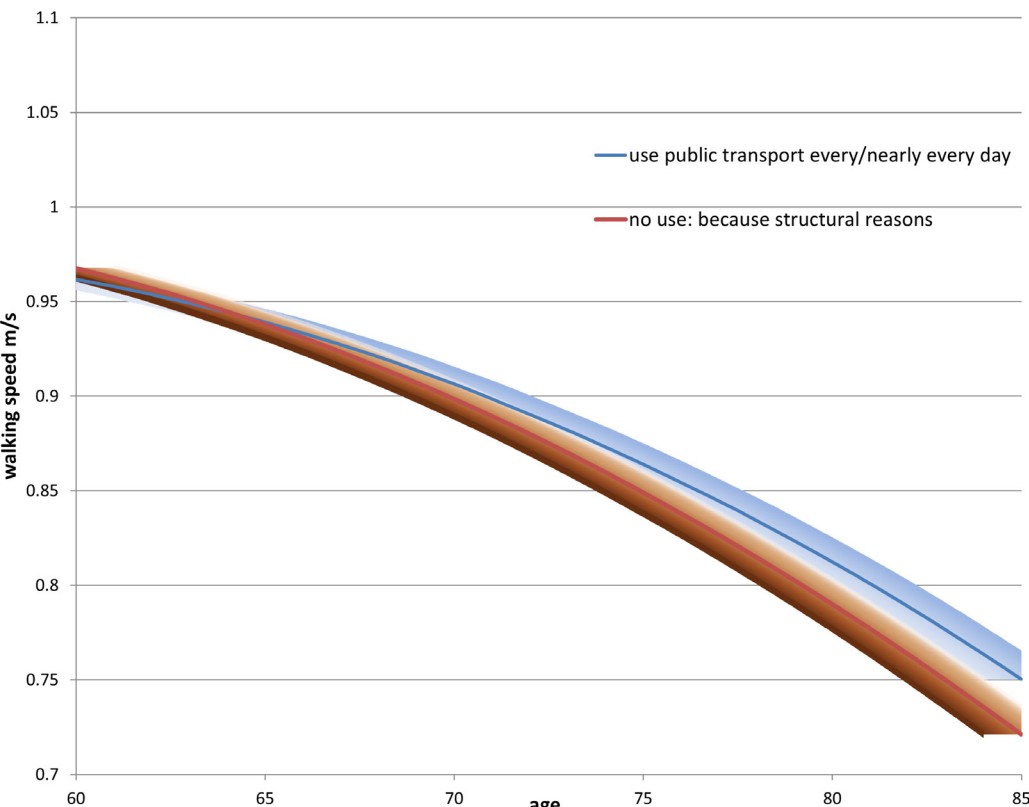

**Figure 1** Predicted decline in walking speed with age by public transport use, comparing ELSA respondents who use public transport every day and those who do not use public transport because of structural reasons. ELSA, English Longitudinal Study of Ageing.

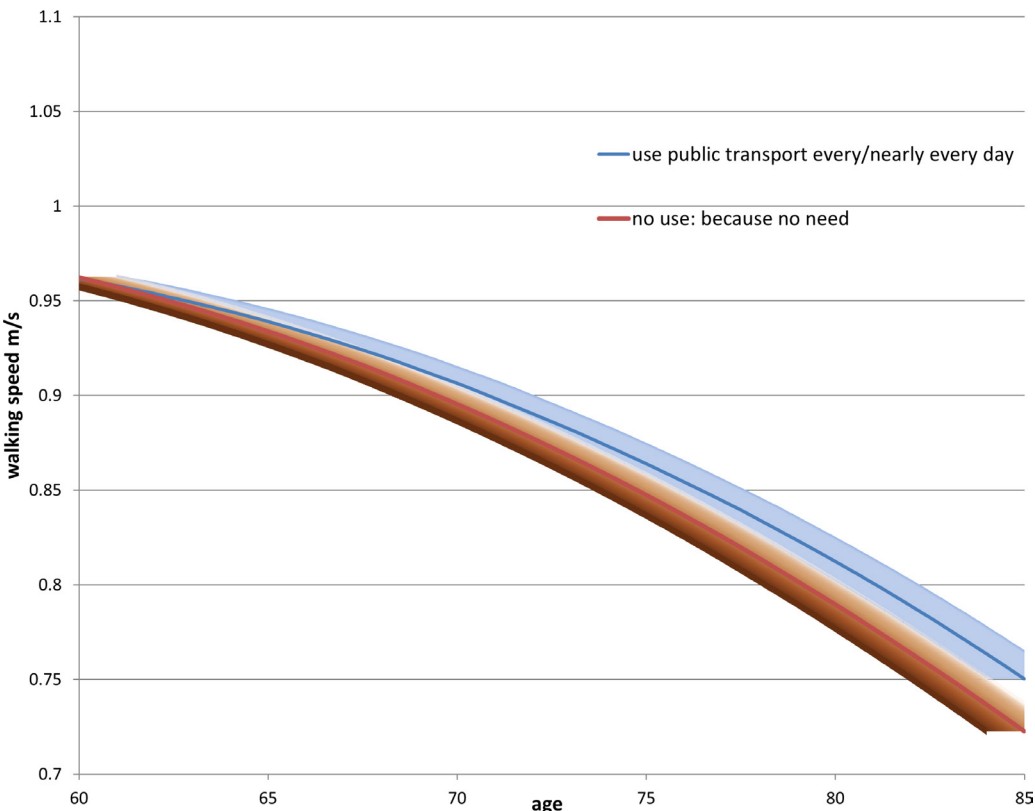

**Figure 2** Predicted decline in walking speed with age by public transport use, comparing ELSA respondents who use public transport every day and those who do not use public transport because they do not need to. ELSA, English Longitudinal Study of Ageing.

The association between public transport use and muscle function deficits was specific to lower limb muscle strength and did not extend to another ageing-related upper body muscle function deficit, grip strength. Frequent use of public transport appears to delay declines in muscles involved in walking, which in turn impacts on walking speed and related physical activity, not other ageing-related muscle function declines. The specificity of the association also suggests that potential confounders related to strength, fitness and health were unlikely to cause the public transport use–walking speed association.

Existing studies have found that use of public transport contributes to better health by increasing physical activity[11 12 18] and reducing obesity.[18 19 23] However, these studies have not examined the reasons why people do not use public transport. Limiting health is potentially a key factor that could confound any association between public transport use and subsequent health. The use of repeated measurements of public transport use (and the underlying reasons for non-use) and walking speed from a large, representative sample of older adults has been useful in taking account of this key confounding factor. Respondents who 'use public transport nearly every day' may be positively health selected. In the analyses, we take account of changes in health conditions in a number of ways. First we control for different health conditions (depression, mobility problems and ADL) that vary across

waves. Second, respondents could select limiting health as the main reason why they could not use public transport—this is the main negative health selection group. Moreover, we found little evidence that respondents with poorer health and slower walking speeds were more likely to report not using public transport because they did not need to or because of structural barriers.

### Limitations

Longitudinal attrition across waves and other missing data may have resulted in a biased sample. The longitudinal sample tended to be older, healthier and more socioeconomically advantaged. We used the wave specific cross-sectional survey weights, which takes account of such predictors of non-response, in order to make the analyses representative of the older population but this may not adequately deal with attrition biases. Furthermore, there may be unobserved factors that cause the association between public transport use and walking speed.

### CONCLUSION

It has become increasingly important for research to show a positive health impact from public transport use, especially among older adults, given cuts to public transport availability in England.[16] Savings to local government from cutting public transport may result in future increased expenditure on ageing-related conditions.

Older adults who do not use public transport frequently are at risk of faster declines in their physical activity, lower limb muscle strength and walking speed compared with those who use public transport every day. This risk was evident not just among older adults who did not use public transport because of health problems, but also among those who did not use public transport because of structural barriers.

**Acknowledgements** The data were made available through the UK Data Archive. ELSA was developed by a team of researchers based at the NatCen Social Research, University College London and the Institute for Fiscal Studies. The data were collected by NatCen Social Research. The funding is provided by the National Institute of Aging in the United States, and a consortium of UK government departments co-ordinated by the Office for National Statistics. The developers and funders of ELSA and the Archive do not bear any responsibility for the analyses or interpretations presented here. PR participated in a Newton Fund/British Council/CONFAP Researcher Links Workshop, 'Indicators for a Healthy City', in Belo Horizonte, Brazil, May 2016, which was coordinated by Professor Jennifer Mindell (UCL) and Professor Waleska Caiaffa (Federal University of Minas Gerais, UFMG).

**Contributors** PR: responsible for study conception, study design, data analysis, and manuscript preparation and revision; had access to all of the data in the study; and takes responsibility for the integrity of the data and the accuracy of the data analysis. TC: contributed to conception and design of the study, data analysis, data interpretation, manuscript preparation and revision. EW: contributed to the design of the study, manuscript revision and provided input on the analysis of the study. All authors: read and approved the final manuscript.

**Funding** PR was supported by a grant from the Economic and Social Research Council [ES/P010075/1]. EW and TC were supported by a grant from the Economic and Social Research Council [ES/J019119/1]. TC was supported by a grant from the Medical Research Council [G1001375].

**Disclaimer** The sponsors had no role in the study design; collection, analysis, and interpretation of the data; writing the report; or the decision to submit the report for publication.

**Competing interests** None declared.

**Patient consent** This study is a secondary data analysis of the English Longitudinal Study of Ageing which is a population-based study not a patient-based study. The National Research Ethics Service approved the study, and all participants gave their informed consent.

**Ethics approval** Ethical approval for all the ELSA waves was granted from NHS Research Ethics Committees under the National Research and Ethics Service (NRES). For further information see:

**Provenance and peer review** Not commissioned; externally peer reviewed.

**Data sharing statement** No additional data are available.

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
