## [Reviewer comments · BMJ Open]

ARTICLE DETAILS

TITLE (PROVISIONAL)	Does public transport use prevent declines in walking speed among older adults living in England? A prospective cohort study
AUTHORS	Rouxel, Patrick; Webb, Elizabeth; Chandola, Tarani

VERSION 1 - REVIEW

REVIEWER	Mark King Centre for Accident Research and Road Safety - Queensland (CARRS-Q) Queensland University of Technology (QUT) Australia
REVIEW RETURNED	09-Jun-2017

GENERAL COMMENTS	This is an important and relevant issue, and the statistical analysis approach is sound. However, in my view there are two apparent confounds in this study that would invalidate the methodology and the interpretation of results. The most important relates to the presumption that self-reports of transport barriers are unrelated to health. As noted by the authors, "Older adults with deteriorating health may be less likely to use public transport and their poor health could determine the decline in walking speed, rather than their lack of public transport use". Using the same line of reasoning, older adults with deteriorating health may report barriers to transport that would not be perceived as barriers by healthier people. There is also the possibility of self-serving rationalisation, whereby it is psychologically more acceptable to attribute non-use of public transport to external barriers rather than to their own physical limitations. If either of these are true, the methodology does not achieve its aim of eliminating the confounding association between health and public transport use. The second potential confound relates to the presumption that "...public transport use is unlikely to have an impact on grip strength" – I'm not sure this is true – if a bus user has to hold on (especially if standing) with repeated tightening and relaxing of grip, might this not increase grip strength? Two additional and related concerns I have are about the practical significance of the results (if accepted), which is linked to the sample definition. While statistically significant, the mean effects on walking speed are very small in a practical sense. I wonder if the need for statistical significance is why the sample was not restricted to the half of the sample with longitudinal data - at the very least I would have preferred to see this group's results reported separately,
--

	especially in relation to RQ3 (trajectories of walking speed). Looking at the description of the variables available, I doubt that my concern about the confounding of self reported transport barriers and health can be addressed, which is why I have recommended rejection rather than revision, although I am happy to be proven wrong.
--	--

REVIEWER	Hans-Werner Wahl Heidelberg University, Germany
REVIEW RETURNED	10-Jun-2017

GENERAL COMMENTS	This is basically an interesting paper, because underscoring that public transport comes along with reduced age-related decline in walking related parameters would be an important public health issues. The current work builds on a long-term observational perspective and a rather large sample substance, which is certainly a good thing. However, there are also open issues which need to be solved to make a really strong case regarding the connection among functional health maintenance and public transport use.  1. I increasingly tend to think that we should generally even in longitudinal research be quite cautious in terms of deriving causal dynamics. What can the author do to rule out that the connection simply works the other way round, hence those better off use more public transport? I expect to make the strongest case possible to underscore that the data support the one-way causal chain as expected by the authors. I have still my doubts here. Of course, there may also be a bidirectional dynamic over time and I regard such a case as much more likely as compared to the assumption that using public transport drives functional health maintenance. Cross-lagged panel analysis or dual change score analysis may help to test the alternative causal pathways. 2. The previous point also has to do with the understanding of the connection between public transport use and walking related parameters. Unfortunately, the paper offers nearly nothing in terms of understanding why such a connection should exist. There are at least two ways to do so. One would be a convincing conceptual framework that suggests why the connection should indeed be substantial. The second possibility, in the ideal case connected with the first option, would be to test such pathways, for example via a mediation analysis. Currently, the reader is left with the fundamental question: Why should such a connection exist? 3. The sample is 50+ and this means that a large portion of the sample is still in the labor force and thus has completely other public transport needs and dynamics. How has this been considered? I am not satisfied to simply control for retired / not retired. I indeed would assume that in the younger part of the sample there is no connection between using public transport and walking capabilities. 4. Please report more details on the drop-out process? How representative is the remaining long-term sample? 5. I was surprised that cognitive ability was not considered as a co-variate? This is a variable with potential connection to public transport use as well as walking parameters. 6. I did not get how the large portions of car drivers have been treated in the analysis and coding of the data. This is likely the majority of the sample and it would be important to underscore that the public transport users operate differently from the car drivers. There is also a considerable group going for combinations (e.g., using the car for
---

	shorter trips, going by train for longer trips). 7. Living alone versus not may also impact on public transport use. Was this considered?
--	--

VERSION 1 – AUTHOR RESPONSE

Reviewer: 1

Reviewer Name: Mark King

This is an important and relevant issue, and the statistical analysis approach is sound. However, in my view there are two apparent confounds in this study that would invalidate the methodology and the interpretation of results.

1.1 The most important relates to the presumption that self-reports of transport barriers are unrelated to health. As noted by the authors, “Older adults with deteriorating health may be less likely to use public transport and their poor health could determine the decline in walking speed, rather than their lack of public transport use”. Using the same line of reasoning, older adults with deteriorating health may report barriers to transport that would not be perceived as barriers by healthier people. There is also the possibility of self-serving rationalisation, whereby it is psychologically more acceptable to attribute non-use of public transport to external barriers rather than to their own physical limitations. If either of these are true, the methodology does not achieve its aim of eliminating the confounding association between health and public transport use.

We now directly investigate evidence for this reasoning in the fixed effects multinomial regression model which investigates the predictors of (change in) use/non-use of public transport (Supplementary Table S5). We report in the results that poorer health and slower walking speed were not associated with respondents becoming more likely to report not using public transport because they did not need to or because of lack of need. We also show that poorer health did predict an increased likelihood of not using public transport because of health problems, and that living in an isolated area predicted an increased likelihood of not using public transport because of structural barriers. We argue that these associations demonstrate some validity for the different reasons people report for not using public transport, and that there is little evidence for other psychological rationalisations for reporting barriers to use/non-use of public transport.

1.2 The second potential confound relates to the presumption that “...public transport use is unlikely to have an impact on grip strength” – I’m not sure this is true – if a bus user has to hold on (especially if standing) with repeated tightening and relaxing of grip, might this not increase grip strength?

We now emphasise that we are using grip strength as an indicator of upper body muscle strength, and contrast the results with new analyses using an indicator of lower body muscle strength (the chair stand test). We would argue that the repeated tightening and relaxing of grip while standing up during a moving bus is unlikely to cause an increase in grip strength among older adults. Firstly, we have examined systematic reviews on the determinants of grip strength and have not found any evidence that public transport use among older adults increases grip strength. Secondly, this assumes older adults are standing up while the bus/tram/train is moving, when many of them could be seated. Thirdly, specific grip and forearm exercises are needed to increase grip strength, and normal everyday activities are very unlikely to increase grip strength.

Finally the fixed effect model of grip strength (Supplementary Table S5) showed there was no association between greater public transport use and stronger grip strength, relative to those who did not use public transport because of lack of need or structural barriers. There is thus no evidence behind the suggestion that older adults who use public transport develop a stronger grip strength

compared to those who do not use public transport.

1.3 Two additional and related concerns I have are about the practical significance of the results (if accepted), which is linked to the sample definition. While statistically significant, the mean effects on walking speed are very small in a practical sense.

While the effect sizes are small, we now mention this explicitly in the discussion and draw the readers' attention to the finding that the predicted levels of walking speed in this cohort of older adults for all categories of public transport use were well below the recommended 1.2 m/s walking speed needed for standard pedestrian crossings. Any increase in the walking speed of older adults through factors such as physical activity and increased public transport use may help them cross the road safely.

1.4 I wonder if the need for statistical significance is why the sample was not restricted to the half of the sample with longitudinal data - at the very least I would have preferred to see this group's results reported separately, especially in relation to RQ3 (trajectories of walking speed).

We now show the results for RQ3 (trajectories of walking speed decline) for the subsample of ELSA longitudinal respondents with at least three waves of walking speed data (Supplementary Table S9). We discuss the very similar coefficients obtained for this subsample compared to the analyses including all ELSA respondents (Supplementary Table S3) in the results section.

1.5 Looking at the description of the variables available, I doubt that my concern about the confounding of self reported transport barriers and health can be addressed, which is why I have recommended rejection rather than revision, although I am happy to be proven wrong.

We would like to thank the reviewer for raising these concerns and we also believe that we have directly addressed these concerns through new analyses and additional results/discussions in the revised manuscript.

Reviewer: 2

Reviewer Name: Hans-Werner Wahl

This is basically an interesting paper, because underscoring that public transport comes along with reduced age-related decline in walking related parameters would be an important public health issues. The current work builds on a long-term observational perspective and a rather large sample substance, which is certainly a good thing. However, there are also open issues which need to be solved to make a really strong case regarding the connection among functional health maintenance and public transport use.

2.1 I increasingly tend to think that we should generally even in longitudinal research be quite cautious in terms of deriving causal dynamics. What can the author do to rule out that the connection simply works the other way round, hence those better off use more public transport? I expect to make the strongest case possible to underscore that the data support the one-way causal chain as expected by the authors. I have still my doubts here. Of course, there may also be a bidirectional dynamic over time and I regard such a case as much more likely as compared to the assumption that using public transport drives functional health maintenance. Cross-lagged panel analysis or dual change score analysis may help to test the alternative causal pathways.

We now include new analyses which look at the reverse association- does walking speed predict changes in use/non-use of public transport? These fixed effects multinomial models (Supplementary Table S5) are substantively and statistically different models than the models predicting changes in walking speed. We show that there is little evidence for the reverse association, that slower walking

speed and poorer health are not associated with increased likelihood of not using public transport because of the lack of need or structural barriers.

2.2 The previous point also has to do with the understanding of the connection between public transport use and walking related parameters. Unfortunately, the paper offers nearly nothing in terms of understanding why such a connection should exist. There are at least two ways to do so. One would be a convincing conceptual framework that suggests why the connection should indeed be substantial. The second possibility, in the ideal case connected with the first option, would be to test such pathways, for example via a mediation analysis. Currently, the reader is left with the fundamental question: Why should such a connection exist?

We apologise that increased physical activity as the mechanism by which public transport use is related to walking speed was not made clearer in the original manuscript. We have now added in explicit reference to the mechanisms of physical activity and maintenance of lower body muscle strength in the Introduction and research questions. The new analyses show that not using public transport use because of structural barriers was associated with increased sedentary physical activity among older respondents (Supplementary Table S8) and also increased risk of lower body muscle strength (through becoming unable to complete the chair stand test in Supplementary Table S7). We also now show how these two factors account for about half of the association between walking speed and not using public transport use because of lack of need/structural barriers (Supplementary Table S4).

2.3 The sample is 50+ and this means that a large portion of the sample is still in the labor force and thus has completely other public transport needs and dynamics. How has this been considered? I am not satisfied to simply control for retired / not retired. I indeed would assume that in the younger part of the sample there is no connection between using public transport and walking capabilities.

The sample was actually aged 60+, not 50+ (the walking speed test was done on those aged 60 and over). Table 1 shows that less than 17% of observations were among those in paid employment. Furthermore, we did not simply distinguish between the retired/not retired in the analyses-employment status was categorised into paid employment, retirement and other, the last category mainly includes people who are economically inactive for health problems (for men) and for family reasons (for women). We would argue that labour market dynamics for a sample with a mean age of 70.5 (reported at the start of the Results) are not perhaps the most important issue in relation to walking speed.

2.4 Please report more details on the drop-out process? How representative is the remaining long-term sample?

We now include more detail on the missing data in the methods section, detail how representative the sample is in Supplementary Table S1 and the relevant methods section, and also discuss the implications of the predictors of missingness for the analyses and whether the weights used in the analyses take account of these predictors of missingness.

2.5 I was surprised that cognitive ability was not considered as a co-variate? This is a variable with potential connection to public transport use as well as walking parameters.

We now include two tests of memory as measures of cognitive ability as covariates in all the analyses. Unfortunately the executive function cognitive test battery was not included in ELSA wave 6, so we could not control for executive function in the analyses.

2.6 I did not get how the large portions of car drivers have been treated in the analysis and coding of the data. This is likely the majority of the sample and it would be important to underscore that the public transport users operate differently from the car drivers. There is also a considerable group going for combinations (e.g., using the car for shorter trips, going by train for longer trips).

Car drivers were not explicitly categorised in the main independent variable- the frequency of use/non-use of public transport- as car drivers can also be public transport users. Respondents with access to a car/van were more likely to report not using public transport due to a lack of need or structural barriers, but over 20% of respondents with access to a car/van used public transport frequently. We also included access to a car/van as a covariate in all the analyses. We tested for an interaction between the public transport use/non use variable and access to a car/van in the analyses, but the interaction term was not significantly different from 0.

2.7 Living alone versus not may also impact on public transport use. Was this considered?

Living alone was measured through cohabiting-partnership status and included as a covariate in all the analyses. The vast majority of respondents not cohabiting with partners were living alone.

VERSION 2 – REVIEW

REVIEWER	Mark King Centre for Accident Research and Road Safety - Queensland (CARRS-Q) Queensland University of Technology (QUT) Australia
REVIEW RETURNED	05-Jul-2017

GENERAL COMMENTS	I am satisfied that the concerns I expressed in my initial review have been addressed. I suggest that the rationale regarding grip strength that was provided in the response letter be briefly referred to or referenced in the text. Also note a typo: under “Analysis” the new text about RQ3 repeats the words “can contribute”.
--

REVIEWER	Hans-Werner Wahl Heidelberg University
REVIEW RETURNED	19-Jul-2017

GENERAL COMMENTS	Although I had acknowledged the value and innovative features of this paper from the beginning, I was a somewhat sceptical based on the original version. However, I have found now all my major points well and convincingly addressed.
--

VERSION 2 – AUTHOR RESPONSE

We have addressed the remaining issues raised by reviewer 1 in the revised manuscript (reviewer 2 did not raise any more issues).

1. "I suggest that the rationale regarding grip strength that was provided in the response letter be briefly referred to or referenced in the text." We now provide a reference in the text about the rationale for analysing grip strength - please see the track changes on page 5.

2. Typo: under "Analysis" the new text about RQ3 repeats the words "can contribute". We have now deleted this repeated phrase.

VERSION 3 – REVIEW

REVIEWER	Mark King Centre for Accident Research and Road Safety - Queensland (CARRS-Q) Queensland University of Technology (QUT) Australia
REVIEW RETURNED	07-Aug-2017

GENERAL COMMENTS	I am satisfied that all comments have been addressed appropriately.
---